# Construction, characterization, and immunization of nanoparticles that display a diverse array of influenza HA trimers

**Alexander A. Cohen, Zhi Yang, Priyanthi N. P. Gnanapragasam, Susan Ou, Kim-Marie A. Dam**, **Haoqing Wang**¤, **Pamela J. Bjorkman**\*

Division of Biology and Biological Engineering, California Institute of Technology, Pasadena, CA, United States of America

¤ Current address: Department of Cellular Physiology and Medicine, Stanford University School of Medicine, Palo Alto, CA, United States of America

\* bjorkman@caltech.edu

## Abstract

Current influenza vaccines do not elicit broadly protective immune responses against multiple strains. New strategies to focus the humoral immune response to conserved regions on influenza antigens are therefore required for recognition by broadly neutralizing antibodies. It has been suggested that B-cells with receptors that recognize conserved epitopes would be preferentially stimulated through avidity effects by mosaic particles presenting multiple forms of a variable antigen. We adapted SpyCatcher-based platforms, AP205 virus-like particles (VLPs) and mi3 nanoparticles (NPs), to covalently co-display SpyTagged hemagglutinin (HA) trimers from group 1 and group 2 influenza A strains. Here we show successful homotypic and heterotypic conjugation of up to 8 different HA trimers to both VLPs and NPs. We characterized the HA-VLPs and HA-NPs by cryo-electron tomography to derive the average number of conjugated HAs and their separation distances on particles, and compared immunizations of mosaic and homotypic particles in wild-type mice. Both types of HA particles elicited strong antibody responses, but the mosaic particles did not consistently elicit broader immune responses than mixtures of homotypic particles. We conclude that covalent attachment of HAs from currently-circulating influenza strains represents a viable alternative to current annual influenza vaccine strategies, but in the absence of further modifications, is unlikely to represent a method for making a universal influenza vaccine.

## Introduction

Each year, influenza virus infections affect 5–30% of the global population, resulting in millions of severe infections and hundreds of thousands of deaths [1]. Yearly epidemics are typically caused by the influenza type A, with a smaller number of infections resulting from type B. Vaccines can minimize the incidence of severe infections; however, they do not offer complete protection and have to be re-administered annually [1, 2]. The lack of complete efficacy of current vaccines can be attributed to several reasons. Mainly, the virus undergoes antigenic drift

**Funding:** This work was supported in part by grants from the National Institute of Allergy and Infectious Diseases of the National Institutes of Health to P.J.B. (1R01AI129784 and P50 AI150464). The funders had no role in study design, data collection and analysis, decision to publish, or preparation of the manuscript.

**Competing interests:** The authors have declared that no competing interests exist.

in which mutations accumulate over time that can allow the virus to evade the humoral immune response [1]. This requires that the vaccine formulation be renewed yearly so that the vaccine strains match the circulating strains as closely as possible. Influenza virus also features high antigenic diversity resulting in predominantly strain-specific antibody responses and making it difficult to recognize conserved regions on the viral antigens [2, 3]. Furthermore, through the mechanism of antigenic shift, the RNA segments from strains of different origins can reassort, resulting in new strains that are typically the cause of global pandemics that can rapidly circulate within an antigenically-naïve population [1]. Altogether, these necessitate the need for a universal flu vaccine that could confer protection against a broad swath of antigenically-distinct strains, thereby eliminating the need for yearly vaccines and offering protection against emergent pandemics.

The antibody response to influenza is primarily directed against the hemagglutinin (HA) and neuraminidase glycoproteins, which appear as a dense array of spikes on the surface of the viral particles [3, 4]. The majority of the neutralizing antibody response is against HA, the most abundant viral surface glycoprotein and the sialic acid-binding receptor that mediates fusion between the viral and host membranes. Influenza HA is a trimer of HA1 and HA2 heterodimers, which can be subdivided into head and stalk domains [3, 4]. The HA head is composed of the middle portion of the HA1 sequence, contains the sialic acid binding site responsible for host cell recognition, and features high variability between different strains/ subtypes. The HA2 subunit along with N- and C-terminal regions of HA1 encodes for the more conserved stalk domain, which contains the fusion peptide involved in viral/host cell membrane fusion [3, 4]. Antibodies against the immunodominant HA head can be strongly neutralizing, but are also strain specific, with the exception of antibodies that recognize the receptor binding site [3, 5, 6]. In contrast, the HA stem is immunosubdominant; however, stem antibodies are often broader, although generally less potent than anti-head antibodies, and can induce antibody-dependent cellular cytotoxicity (ADCC) responses [3, 7–9]. Within the past 10 years, broadly neutralizing antibodies (bNAbs) against influenza that target conserved HA stem epitopes have been discovered, but these antibodies have thus far been difficult to elicit [10]. It is generally believed that a broadly protective or "universal" vaccine would require the induction of anti-HA stem antibodies. As a result, there have been numerous attempts to refocus the immune response to these conserved epitopes [11–16].

One strategy to redirect the antibody response towards invariant epitopes was co-displaying influenza HAs from different strains on nanoparticles [16]. The rationale was to display HAs from several strains on a multimerized platform, such that any two adjacent HAs have a low probability of being identical, thereby giving a competitive advantage to B-cells with B-cell receptors (BCRs) that use avidity effects to recognize conserved epitopes shared between different strains. By contrast, BCRs that recognize strain-specific epitopes could not use avidity effects to bind adjacent HAs, thus would be less likely to be activated [16]. In this study, monomeric HA receptor binding domain (RBD) sequences were fused to an engineered ferritin subunit to create self-assembling particles displaying up to eight different RBD sequences derived from H1N1 strains at 24 total positions [16]. The elicited humoral immune response in injected mice featured high breadth and potency against a panel of diverse H1N1 strains, which was most apparent the larger the number of HAs that were co-displayed, such that the simultaneous display of 8 different HAs elicited the greatest breadth in comparison with immunization of a cocktail of 8 different homotypic HA nanoparticles. Sorting and isolation of memory B-cells that were positive for HAs from two different strains further supported the use of this strategy for inducing cross-reactive B-cells. A logical follow up to this study would be to co-display HA ectodomain trimers including the stalk and head regions instead of RBD head domain monomers, with the hope of eliciting antibody lineages with increased breadth.

We reasoned that co-display of multiple HA trimers on a nanoparticle would be facilitated by a system in which soluble HA trimers could be covalently attached to a protein nanoparticle, thereby avoiding potential folding problems created by genetically fusing protomers from a trimer to a nanoparticle subunit. Numerous NP platforms and coupling strategies have been explored for vaccine design [17]. The "plug and display" strategy involves the use of virus-like particles (VLPs) or nanoparticles (NPs) fused to a SpyCatcher protein that is covalently conjugated to a purified antigen tagged with a short (13-residue) SpyTag [18, 19]. The conjugation involves the formation of an isopeptide bond between a lysine from the SpyCatcher protein and an aspartate from the SpyTag [20]. An advantage of the SpyCatcher-SpyTag system is that it allows for the spontaneous irreversible conjugation of a purified antigen with native-like post-translational modifications to a scaffold via an incubation of the antigen and scaffold proteins under physiological conditions. Available SpyCatcher protein scaffolds are highly versatile, coming in different forms that range from a bacteriophage AP205 T = 3 icosahedral particle (180 SpyCatchers) to a designed dodecahedral NP called mi3 (60 SpyCatchers) [18, 19]. We recently used AP205 SpyCatcher-VLPs to display SpyTagged trimeric HIV-1 Env immunogens and demonstrated priming in immunized mice and non-human primates of B-cells carrying receptors displaying characteristics of V3-glycan patch-targeting HIV-1 bNAbs [21].

Here we describe the use of bacteriophage AP205-based SpyCatcher VLPs and engineered particle mi3-based SpyCatcher NPs [18, 19] to display a diverse array of HA ectodomain trimers from group 1 and group 2 influenza A strains. Successful conjugation was demonstrated by size-exclusion chromatography (SEC), SDS-PAGE, and electron microscopy (EM), with up to eight different HA trimers successfully conjugated to mosaic mi3 particles. Our results demonstrated that SpyCatcher-VLPs and SpyCatcher-NPs can be easily used to stably display at least 8 different trimeric antigens and that AP205-HA and mi3-HA particles produced strong immune responses in mice.

## Materials and methods

### Expression and purification of soluble HA trimers

HA ectodomain trimers were expressed as shown schematically in Fig 1A with a C-terminal foldon trimerization domain, 13-residue SpyTag [20], and a 6x-His (modified from HA constructs in [22] to include a SpyTag). Genes corresponding to the modified HA1-HA2 sequences (residues 1–504 H3 numbering) from A/Aichi/02/1968 (Aichi; H3), A/Shanghai/1/2013 (SH13; H7), A/Jiangxi-Donghu/346/2013 JX346; H10), A/swine/HuBei/06/2009 (HB09; H4), A/California/04/2009 (CA09; H1), A/Vietnam/1203/ 2004 (Viet04; H5), A/Japan/305/1957 (JP57; H2), and A/guinea fowl/Hong Kong/1999 (WF10; H9N2) were subcloned into a pTT5 expression vector. Genes encoding SpyTagged HAs with a Y98F mutation (H3 numbering) were constructed using site directed mutagenesis. HA ectodomain trimer constructs were expressed by transient transfection using the Expi293 Expression System (ThermoFisher), and soluble HA trimers were purified from transfected cell supernatants by standard Ni-NTA chromatography using a prepacked HisTrap™ HP column (GE Healthcare) and SEC using a HiLoad® 16/600 Superdex® 200 column (GE Healthcare). Proteins were concentrated using an Amicon Ultra 15 mL 30K concentrator (MilliporeSigma) and stored at 4°C in 20 mM Tris pH 8.0, 150 mM NaCl, 0.02% NaN$_3$ (TBS buffer).

HA ectodomain trimers for ELISAs were expressed as above without the 13-residue SpyTag or the Y98F substitution. Additional strains only used for ELISA include: A/shearwater/West Australia/2576/79 (WA79; H15) and A/flat-faced bat/Peru/033/2010 (Pe10; H18). The CA09-miniHA construct (construct #4900) [11] was subcloned into a pTT5 mammalian

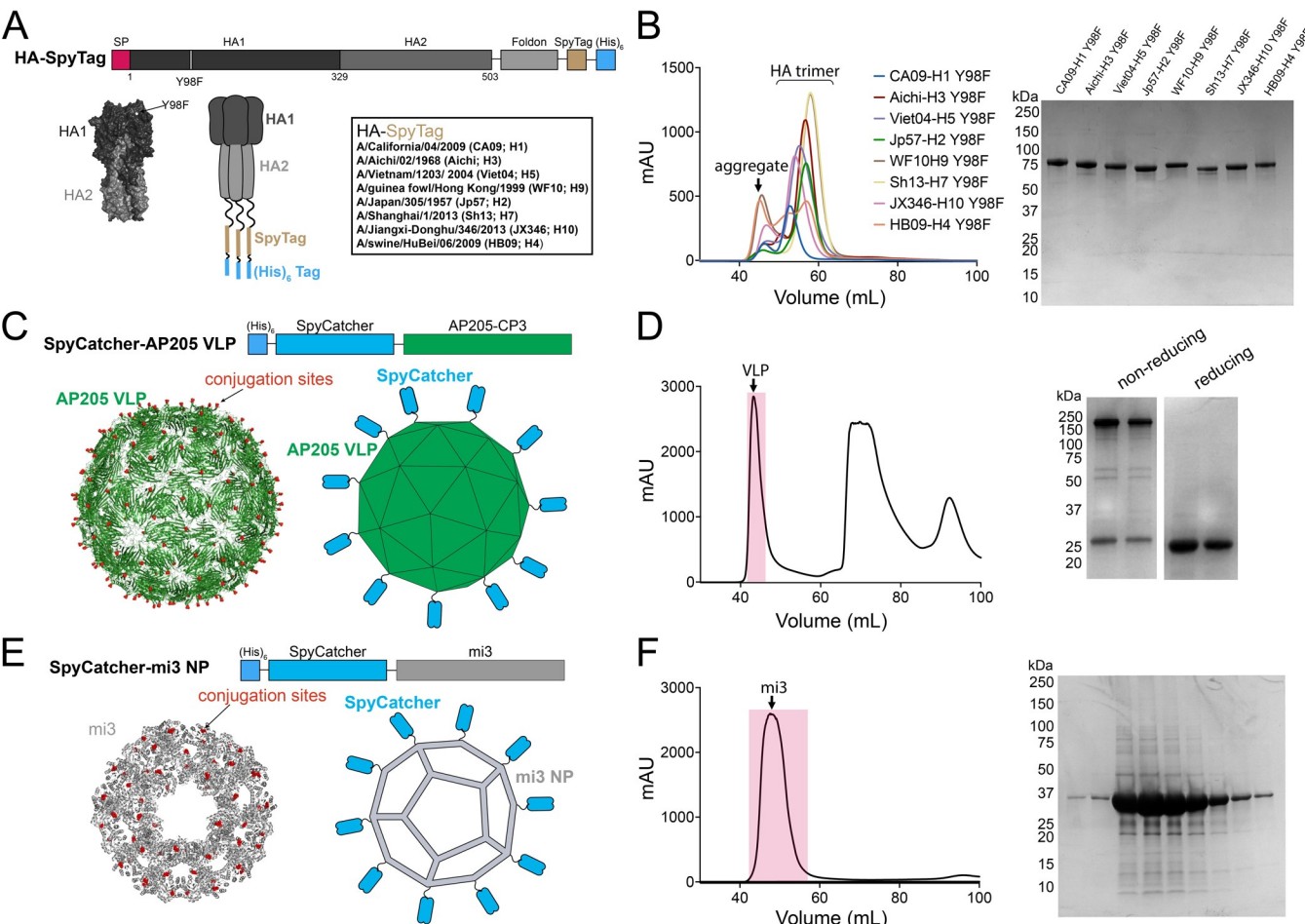

**Fig 1. Design and characterization of SpyTagged HAs, SpyCatcher VLPs, and SpyCatcher-NPs. A**. Top: Schematic of the SpyTagged HA construct (SP = signal peptide). The HA2 ectodomain is followed by a foldon trimerization domain from T4 fibritin, a 13-residue SpyTag, and a 6x-His tag. Amino acids are numbered according to the H3 nomenclature. Horizontal lines represent Gly$_4$Ser linkers. Bottom: Surface representation of an HA trimer structure (PDB 3VUN), schematic of a SpyTagged HA, and list of influenza strains from which SpyTagged HAs were derived. **B**. SEC profiles and reducing SDS-PAGE analysis of 8 purified SpyTagged HAs that contain Y98F substitution. **C**. SpyCatcher-AP205 VLPs. Top: Schematic of construct. Bottom: EM structure of T = 3 AP205 particle (PDB 5FS3) [23] with the locations of SpyCatcher fusion sites indicated by red dots (left) and schematic SpyCatcher-VLP (right). **D**. Purification of SpyCatcher-VLPs. Left: SEC profile with peak representing properly-assembled VLPs indicated. Right: Reducing and non-reducing SDS-PAGE of two fractions corresponding to the VLP fractions in red on the SEC trace. **E**. SpyCatcher-mi3 NPs. Top: schematic of construct. Bottom: Cryo-EM structure of I3-01 particle related to mi3 [24] with the locations of SpyCatcher fusion sites indicated by red dots (left) and schematic SpyCatcher-NP (right). **F**. Purification of SpyCatcher-NPs. Left: SEC profile. Right: Reducing SDS-PAGE of fractions corresponding to the mi3 fractions shaded in red on the SEC trace.

expression vector with a 6x-His tag and expressed and purified as described for the HA ectodomain trimers. For flow cytometry experiments, an Avi-tag was inserted after the C-termini of the Aichi, Viet04, and CA09 HAs with the Y98F substitution. Avi-tagged HAs were expressed and purified as described above and biotinylated using the Biotin ligase kit (Avidity) according to the manufacturer's protocol. Biotinylated CA09-HA, Aichi-HA and Viet04-HA were incubated with eBioscience™ Streptavidin APC, Streptavidin PE-eFluor™ 610, or Streptavidin PE (ThermoFisher) overnight at 4°C at a 1:1 molar ratio of HA trimer to streptavidin subunit.

## Expression of SpyCatcher-VLPs and SpyCatcher NPs

pGEM-SpyCatcher-AP205-CP3 for expression of SpyCatcher-VLPs was the kind gift of Dr. Mark Howarth (Oxford University). pGEN SpyCatcher AP205-CP3 was transformed into

OverExpress™ C41(DE3) *E.coli* (Sigma). Single colonies were picked and inoculated into a 2xYT (Sigma) overnight starter culture and then grown in 1L 2xYT media with shaking at 220 rpm at 37°C until OD 0.5 ($A_{600}$), after which they were induced with 0.42 mM IPTG and grown for 5 hours at 30°C. Cultures were harvested and pellets were frozen in lysis buffer (20mM Tris-HCl pH 7.8, 150mM NaCl, 0.1% Tween 20, 75 mM imidazole). For producing VLPs for conjugation, pellets were thawed and lysed with a cell disruptor in the presence of 2.0 mM PMSF (Sigma). The lysate was spun at 21,000xg for 30 min, filtered with a 0.2 μm filter, and VLPs were isolated by Ni-NTA chromatography using a prepacked HisTrap™ HP column (GE Healthcare). Spy-Catcher VLPs were eluted with 2.0 M imidazole, 50 mM glycine, 25 mM sodium citrate, 0.1% Tween 20, pH 8.5. Eluted VLPs were concentrated using an Amicon Ultra 15 mL 30K concentrator (MilliporeSigma) and further purified by SEC using a HiLoad® 16/600 Superdex® 200 (GE Healthcare) column equilibrated with 500 mM glycine pH 8.0, 250 mM sodium citrate, 1% Tween 20. VLPs were then stored at 4°C and used for up to 1 month for conjugations. Spy-Catcher-VLPs precipitated out of solution over time and before conjugations, they were either filtered with a 0.2 μm filter or spun down at 21,000g for 10 min.

The pET28a His6-SpyCatcher-mi3 gene (Addgene) was transformed into BL21 (DE3)-RIPL *E.coli* (Agilent). Single colonies were picked and inoculated into an LB overnight starter culture, and grown in 1L LB media until OD 0.8 ($A_{600\ nm}$) with shaking at 220 rpm at 37°C, after which they were induced with 0.5 mM IPTG and grown for 16–20 hours at 20°C. Cultures were then harvested and pellets were frozen in lysis buffer (250 mM Tris-HCl pH 8.0, 150 mM NaCl, 50 mM imidazole, 0.02% $NaN_3$). For producing NPs for conjugation, pellets were thawed and lysed with a cell disruptor in the presence of 2.0 mM PMSF (Sigma), and the lysate was spun at 21,000xg for 30 min, filtered with a 0.2 μm filter, and NPs were isolated by Ni-NTA chromatography using a prepacked HisTrap™ HP column (GE Healthcare), and eluting with 2.0 M imidazole, 20 mM Tris-HCl pH 8.0, 150 mM NaCl, 0.02% $NaN_3$. Eluted NPs were concentrated using an Amicon Ultra 15 mL 30K concentrator (MilliporeSigma) and further purified by SEC using a HiLoad® 16/600 Superdex® 200 (GE Healthcare) column equilibrated with 25 mM Tris-HCl pH 8.0, 150 mM NaCl, 0.02% $NaN_3$. NPs were then stored at 4°C and used for up to 1 month for conjugations. SpyCatcher-NPs precipitated out of solution over time, and before use for conjugations, they were either filtered with a 0.2 μm filter or spun down at 21,000g for 10 min.

## Preparation of HA-VLPs and HA-NPs

Purified SpyCatcher-VLPs or SpyCatcher-NPs were incubated with a 1.2-fold molar excess (HA protomer to VLP or NP subunit) of purified SpyTagged HA (either a single HA for making homotypic particles or an equimolar mixture of two or more HAs for making mosaic particles) at room temperature in TBS (25 mM Tris-HCl pH 8.0, 150 mM NaCl, 0.02% $NaN_3$) overnight. Conjugated VLPs or NPs were then separated from free HA trimers by SEC on a Superose 6 10/300 (GE Healthcare) column equilibrated with PBS (20 mM sodium phosphate pH 7.5, 150 mM NaCl). Fractions corresponding to conjugated VLPs or NPs were collected and analyzed via SDS-PAGE. Concentrations were determined using a Bio-Rad Protein Assay. For stability studies, mosaic NP preps were stored for a month at 4°C and then analyzed via SEC using a Superose 6 10/300 (GE Healthcare) column equilibrated with PBS (20 mM sodium phosphate pH 7.5, 150 mM NaCl).

## EM

HA-conjugated and unconjugated VLPs and NPs were compared by negative-stain EM. Ultra-thin, holey carbon-coated, 400 mesh Cu grids (Ted Pella, Inc.) were glow discharged for 60 s at

15 mA. A 3-μl aliquot of SEC-purified HA-VLPs and HA-NPs diluted to approximately 40–100 ug/ml were applied to the grids for 60 s, and then negatively stained with 2% (w/v) uranyl acetate for 30 s. Data were collected with a FEI Tecnai T12 transmission electron microscope at 120 keV at 42,000x magnification.

SEC-purified HA-VLPs and HA-NPs were prepared on grids for cryo-ET using a Mark IV Vitrobot (ThermoFisher Scientific) operated at 21˚C and 100% humidity. 3.1 μL of sample was mixed with 1 μL of 10 nm colloidal gold beads (Sigma-Aldrich) as fiducial markers and then applied to 300 mesh Quantifoil R2/2 grids, blotted for 3.5 s, and plunge-frozen in liquid ethane surrounded by liquid nitrogen. Cryo-ET was performed on a 300kV Titan Krios transmission electron microscope (ThermoFisher Scientific) equipped with a Gatan energy filter (slit width 20 eV) operating at a nominal 33,000x magnification. For HA-VLPs, tilt series were collected on a K2 direct electron detector (Gatan) with a pixel size of 2.23 Å•pixel$^{-1}$ using SerialEM software [25], a -3 to -6 μm defocus range, and a total of 98 e$^-$•Å$^{-2}$ per tilt series. For HA-NPs, tilt series were recorded in counting mode on a K3 direct electron detector (Gatan) with a pixel size of 2.68 Å•pixel$^{-1}$ using SerialEM [25], a -4 to -5 μm defocus range, and a total dose of ~140 e$^-$•Å$^{-2}$ per tilt series. For both data collections, tilt series images were collected using a dose-symmetric tilt scheme [26] ranging from -60˚ to 60˚ with 2˚ and 3˚ intervals for HA-VLPs and HA-NPs, respectively. Images were aligned and reconstructed using IMOD [27, 28].

## Immunizations

All animal experiments were carried out in 4–6 week old female Balb/c mice obtained from Charles River Laboratories. The immunizations with HA-VLPs and HA-NPs in Figs 4A and 5A, respectively, were done in Balb/c mice (n = 4 in each group) through intraperitoneal (ip) injections of 20 μg of antigen in 200 μL of 50% v/v of adjuvant (Sigma Adjuvant System®). For experiments in Fig 4A, mice were immunized on Day 0 and boosted on Day 14. Animals were bled weekly via tail veins. For animals in Fig 5A, mice were also boosted on Day 37. Mice were euthanized 2 weeks later (Day 49, 51), bled through cardiac puncture, and spleens were harvested. For Fig 7A, mice (n = 5 except for mi3- or VLP-immunized mice, where n = 2) were immunized with the indicated immunogen in 100 μL of 50% v/v of AddaVax™ adjuvant (Invivogen) and boosted with adjuvant on Day 14, 28, and 168. This adjuvant was chosen in order to compare with previous experiments [16, 29]. Mice were euthanized 2 weeks after the final boost (Day 182,183), bled through cardiac puncture, and spleens were harvested. All blood samples were allowed to clot at room temperature in MiniCollect® Serum and Plasma Tubes (Greiner), and then serum was harvested, frozen in liquid nitrogen, and stored at -80˚C until use. All of the animal experiments were performed using experimental protocols approved by the Institutional Animal Care and Use Committee (IACUC), California Institute of Technology (Protocol IA19-1725). Animals were euthanized at the end of the experiment and spleen tissue will be harvested for in vitro analysis to look at the memory B-cell response. Animals were euthanized with CO2 inhalation, then prolonged exposure and confirmation of lack of heartbeat and/respiratory rate.

## ELISAs

Nunc® MaxiSorp™ 384-well plates (Sigma) were coated with 10 μg/ml of a purified HA (without a SpyTag) in 0.1 M NaHCO$_3$ pH 9.8 and stored overnight at 4˚C. Plates were blocked with 3% bovine serum albumin (BSA) in TBS-T (TBS with 0.1% Tween 20) for 1 hr at room temperature. Plates were washed with TBS-T after each step. Serum was diluted 1:100 and then serially diluted by 4-fold with TBS-T/3% BSA and added for 3 hr at room temperature. A 1:50,000 dilution of secondary HRP-conjugated goat anti-mouse IgG (Abcam) was added for 1

hr at room temperature. Plates were developed using SuperSignal™ ELISA Femto Maximum Sensitivity Substrate (ThermoFisher) and read at 425 nm. Curves were plotted and integrated to obtain the area under the curve (AUC) using Graphpad Prism 8.3. Statistical differences of AUC titers between groups were calculated using Tukey's multiple comparison test via Graphpad Prism 8.3.

## In vitro neutralization assays

Neutralization assays were conducted using live PB1flank-eGFP virus for BSL 2 strains A/Aichi/02/1968 (X31; H3N2), A/California/04/2009 (CA09; H1N1), A/Texas/36/1991 (TX91; H1N1), and A/Wisconsin/67/2005 (WI05; H3N2) as described [30] using reagents kindly provided by Dr. Jesse Bloom (Fred Hutchinson). Plasma was set at a top dilution of 1:200 (for Fig 4C) or 1:100 (For Fig 5C) and serially diluted 5-fold (for Fig 4C) or 4-fold (For Fig 5C) for a total of 8 dilutions. Pseudovirus assays were conducted as described [31] for BSL 3 strains A/Shanghai/1/2013 (SH13; H7), A/Jiangxi-Donghu/346/2013 (JX346; H10), A/Vietnam/1203/2004 (Viet04-H5), and A/Netherlands/219/2003 (NL03-H7). Plasma was set at a top dilution of 1:200 (for Fig 4C) or 1:100 (For Fig 5C) and serially diluted 4-fold for a total of 8 dilutions. Neutralization data were plotted, curves were fit, and $ID_{50}$ values were calculated using Antibody Database [32]. Reported $IC_{50}$s are geometric means, which are suitable for data sets covering multiple orders of magnitude [33]. Statistical differences of $ID_{50}$ titers between groups were calculated using Tukey's multiple comparison test via Graphpad Prism 8.3. Correlation between neutralization $ID_{50}$s and ELISA AUC titers were calculated using the Pearson correlation function on Graphpad Prism 8.3.

## Flow cytometry

Single cell suspensions were prepared from immunized mouse spleens by mechanical dissociation using the back of a syringe plunger. Cell suspensions in 70 μm cell strainers were washed in cold RPMI 1640 media and treated with ACK lysing buffer (Gibco) to lyse red blood cells. The resulting white blood cell preparation was resuspended in RPMI 1640 MACS and enriched for memory B-cells using the negative selection portion of the protocol in a mouse Memory B-cell Isolation Kit (Miltenyi). For the experiment in Fig 6A, enriched splenocytes were then stained with the following monoclonal antibodies and reagents: CD4-APC-eFluor 780 (clone: RM4-5), F4/80-APC-eFluor 780 (clone: BM8), CD8a-APC-eFluor 780 (clone: 53–6.7), Ly-6G-APC-eFluor 780 (clone: RB6-8C5), IgM-PerCP-eFluor 710 (clone: II/41) (eBioscience), CD19-FITC (clone: 6D5) (Biolegend), IgG1 BV421 (clone: X40), IgG2 BV421 (clone: R19-15) (BD Bioscience), and CA09-HA-APC, Aichi-HA-PEeflour610 and Viet04-HA-PE (prepared as described above). Cell viability was analyzed with Ghost Dye™ Violet 510 (Tonbo). For the experiment in Fig 8A, enriched splenocytes were stained with the following monoclonal antibodies and reagents: CD4-APC-eFluor 780 (clone: RM4-5), F4/80-APC-eFluor 780 (clone: BM8), CD8a-APC-eFluor 780 (clone: 53–6.7), Ly-6G-APC-eFluor 780 (clone: RB6-8C5), IgM- APC-eFluor 780 (clone: II/41) (Thermo-fisher Scientific), CD19-FITC (clone: 6D5) (Biolegend), IgG1 BV421 (clone: X40), IgG2 BV421 (clone: R19-15) (BD Bioscience), and CA09-HA-APC, Viet04-HA-PE for looking at CA09+Viet04+ B-cells, CA09-HA-APC and Aichi-HA-PE for looking at CA09+Aichi+ B-cells, or Sh13-HA-APC and Aichi-HA-PE for looking at Sh13+Aichi+ B-cells (all HA probes were prepared as described above). Cell viability was analyzed with Ghost Dye™ Violet 510 (Tonbo). Splenocytes were incubated for 30 min at 4°C in the dark and then washed twice with staining buffer (HBSS, 50 mM HEPES pH 7.4, 2.5 mg/ml BSA, 50 ug/ml DNAse, 1 mM MgCl$_2$). Stained cells were then analyzed with a SY3200 Cell Sorter (Sony) configured to detect 9 fluorochromes. 500,000–

1,000,000 events were collected per sample and analyzed via FlowJo software (TreeStar). Statistical differences of antigen-specific B-cell populations between groups were calculated using Tukey's multiple comparison test via Graphpad Prism 8.3. Correlation between percentage of antigen-specific B-cells and ELISA AUC titers were calculated using the Pearson correlation function on Graphpad Prism 8.3.

## Results and discussion

### Construction of HA-VLPs and HA-NPs

We adapted the AP205 SpyCatcher-VLP platform that we had previously used to conjugate a trimeric HIV-1 immunogen [21] as a way to increase the intrinsic immunogenicity of HA, mask undesired epitopes located at the bottom of the HA trimer, and attach different HA trimers to the same particle. We reasoned that the SpyCatcher-VLP platform could be used to display more than one HA by incubating with equimolar amounts of different SpyTagged HAs. Although the SpyTagged HAs would be conjugated at random to available SpyCatcher proteins, there should be no advantage for the conjugation of one HA over another since they all contained the same SpyTag.

We first expressed and purified SpyTagged soluble HA trimers derived from 8 different influenza strains from group 1 and group 2 influenza A viruses (Fig 1A). The constructs for each HA protomer contained HA1 and the HA2 ectodomain (residue 1–503 H3 numbering) linked to a C-terminal foldon trimerization domain, a SpyTag and a 6x-His tag (Fig 1A). The HAs for the SpyCatcher-mi3 conjugations included the sialic acid binding knockout mutation Y98F (except for the SpyTagged HAs used to conjugate the SpyCatcher-VLPs). SpyTagged HAs including the Y98F substitution purified from the supernatants of transiently-transfected mammalian cells were verified to form monodisperse and well-behaved trimers by SEC and SDS-PAGE (Fig 1B).

We used SpyCatcher-AP205 VLPs and SpyCatcher-mi3 NPs as conjugation platforms for multivalent display of HA trimers. AP205-SpyCatcher VLPs are icosahedral capsids (T = 3 symmetry) with 180 copies total, therefore 180 SpyCatchers were available for conjugation (Fig 1C). AP205-SpyCatcher VLPs were expressed in *E. coli* and purified via Ni-NTA affinity chromatography followed by SEC (Fig 1D) [18]. The Spycatcher-AP205-VLPs eluted near the void volume as a single monodisperse peak. SpyCatcher-mi3-NPs are an engineered dodecameric scaffold with 60 total subunits, therefore 60 conjugation sites [19] (Fig 1E). The SpyCatcher-mi3s were also expressed in *E. coli*, purified by Ni-NTA affinity chromatography followed by SEC, and analyzed with reducing SDS-PAGE (Fig 1F).

Using the SpyCatcher-AP205 VLPs, we first evaluated coupling of two recombinant HAs, A/California/04/09 H1 (CA09-HA) and A/Aichi/02/1968 H3 (Aichi-HA) (chosen to represent two strains that would normally be present in an annual influenza vaccine [2, 3]). Conjugations of the AP205-Spycatcher VLPs were carried out by room temperature incubation with CA09-HA, Aichi-HA, or an equimolar mixture of both HAs in a 1.2 molar excess to the VLPs (HA protomer to VLP subunit) to prepare CA09-, Aichi-, and mosaic-2 VLPs, respectively (Fig 2A). VLP-conjugated HA trimers were separated from free trimers by SEC (Fig 2B), and successful conjugation of the SpyCatcher VLPs to Aichi-HA, CA09-HA, and both HAs was verified by a shift in apparent molecular weight (from 75 kDa to 100 kDa) detected by SDS-PAGE for HAs conjugated to the VLP subunits (Fig 2B).

Because VLPs conjugated with more than two different HAs tended to precipitate out of solution, we switched to the SpyCatcher-mi3 NP platform, which is similarly immunogenic as the AP205 platform, but has been shown to exhibit improved yields, stability, and uniformity [19]. In addition, we modified the SpyTagged HAs to include a receptor binding site mutation,

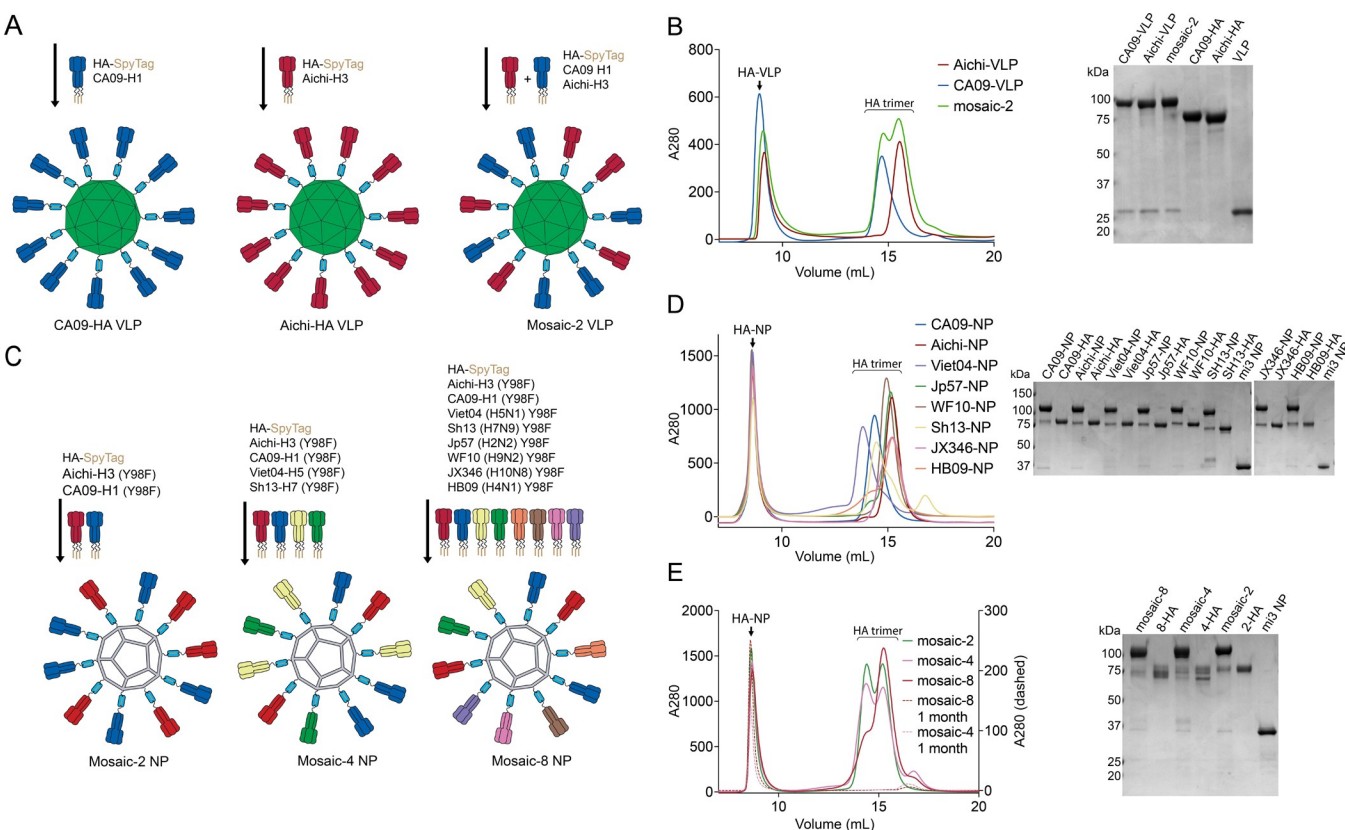

**Fig 2. Conjugation of SpyCatcher-VLPs and -NPs. A**. SpyCatcher-AP205-VLP conjugations with SpyTagged-HA trimers (without the Y98F substitution). **B**. Purification of conjugated SpyCatcher-VLPs. Left: SEC separation of conjugated VLPs from free HA trimers. Right: Reducing SDS-PAGE analysis of VLPs and purified HAs. **C**. SpyCatcher-mi3 NP conjugations with SpyTagged-HA Y98F trimers. **D**. Purification of homotypic SpyCatcher-NPs. Left: SEC separation of conjugated NPs from free HA trimers. Right: Reducing SDS-PAGE analysis of NPs and purified HAs. **E**. Purification of heterotypic mosaic NPs. Left: SEC separation of conjugated NPs from free HA trimers, including SEC profile of purified conjugated NPs after one month storage at 4 ˚C. Right: Reducing SDS-PAGE analysis of NPs and purified HAs.

Y98F (H3 numbering), to abolish sialic acid binding [34] that could result in interactions of aggregation of HAs on neighboring particles. Starting with 8 HAs from influenza group 1 and group 2 strains (Fig 1A) with pandemic potential [35], we made mosaic-2, mosaic-4 and mosaic-8 NPs (each with an equal representation of group 1 and group 2 strains) and the 8 corresponding homotypic HA-conjugated NPs (Fig 2C). Homotypic and mosaic HA-mi3s were purified via SEC, and conjugation was verified by a shift in apparent molecular weight (from 75 kDa to >100 kDa) detected by SDS-PAGE for HAs conjugated to the mi3 subunits (Fig 2D and 2E). To assess stability of the conjugated NPs, mosaic NP samples stored for one month at 4˚C were analyzed for degradation by SEC, revealing little to no free HA trimer for both the mosaic-4 and mosaic-8 NPs (Fig 2E).

## EM characterization of HA-VLPs and HA-NPs

Negative-stain EM revealed increased diameters for conjugated VLPs and NPs compared with their unconjugated counterparts (Fig 3A). HA-conjugated VLPs were also examined by single-particle cryo-EM. 2D class averages of mosaic-2 VLPs showed ordered density for the AP205 VLP, but blurred densities for attached HAs (Fig 3B), suggesting variability in trimer orientations with respect to the VLP surface.

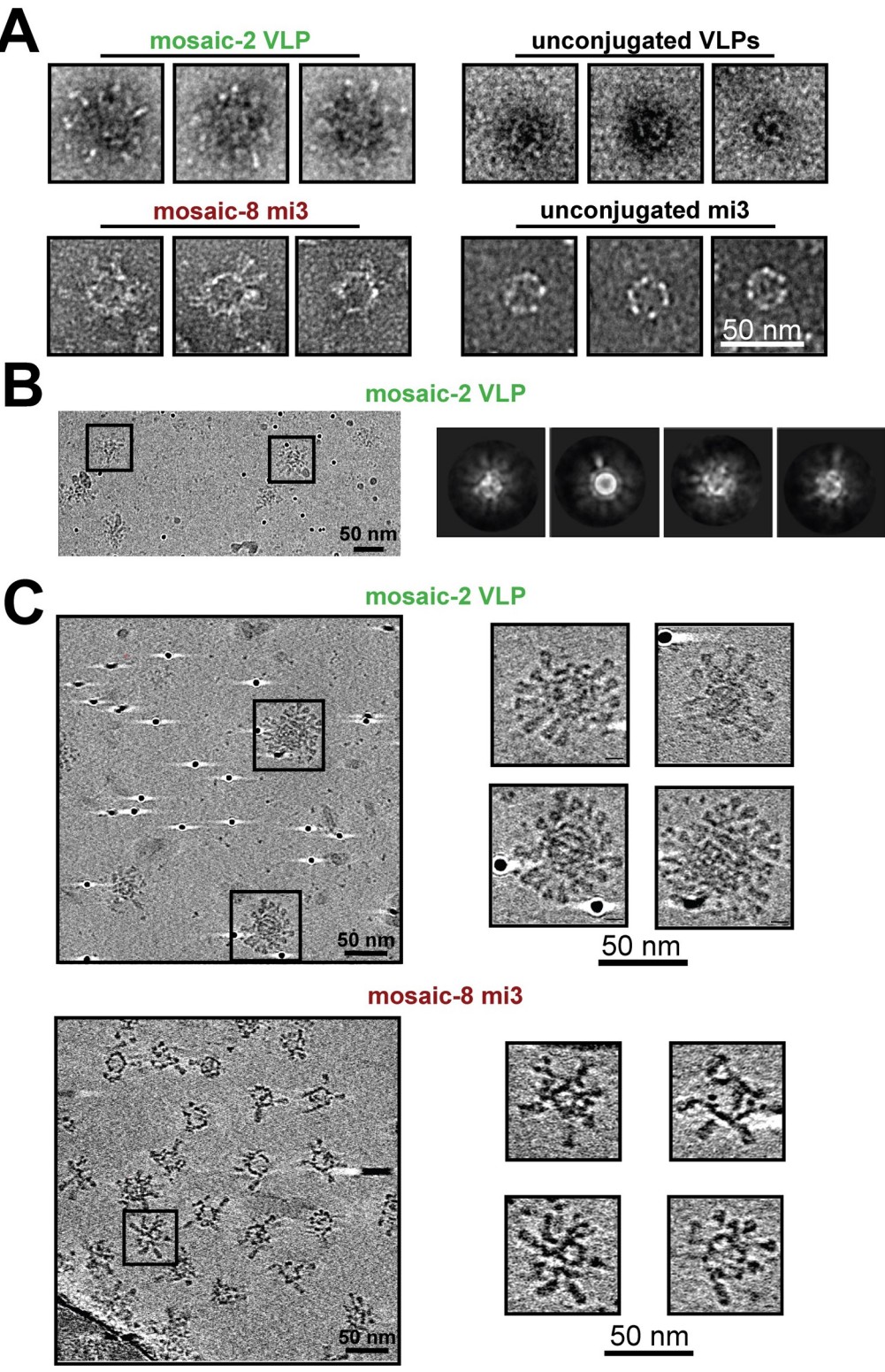

**Fig 3. EM of conjugated VLPs and NPs.** Scale bars shown apply to all images in each panel. **A.** Negative-stain EM of HA-conjugated VLPs and mi3 NPs compared with unconjugated counterparts. **B.** Cryo-EM micrograph of HA-VLP sample (left) and representative 2D class averages (right). Densities for HA trimers are blurry in the class averages, likely because the trimers occupy different positions on individual particles. **C.** Cryo-ET imaging of HA particles. Computationally-derived tomographic slices of HA-VLP (top panels; 2.78 nm slices) and HA-NP (bottom panels; 3.21 nm slices). Slices derived from the widest portions of representative particles are shown to the right in each panel.

Since the HA trimer densities could not be reliably interpreted by single-particle cryo-EM, we used cryo-ET to derive 3D reconstructions of individual HA-conjugated VLPs and NPs. Tomograms (S1 and S2 Movies) showed particles with average diameters of 60 nm (HA-VLPs) and 50 nm (HA-NPs) and revealed densities for individual HA trimers on VLPs and NPs. The trimers were separated by distances of ~7–10 nm and ~12–15 nm for VLPs and NPs, respectively (measured between the head regions of trimer axes on adjacent HAs). To estimate the number of conjugated HA trimers, we counted HA densities in ~3 nm tomographic slices of individual HA-VLPs and HA-NPs at their widest diameters, where the symmetries of each type of particle predicted a maximum of 20 potential attachment sites. We found 9–16 HA densities for conjugated VLPs and 6–8 densities for conjugated NPs, corresponding to occupancies of 45–80% (VLPs) and 30–40% (NPs). Since AP205 VLPs and mi3 NPs contain 180 or 60 SpyCatcher domains, respectively, this translates to ~81–144 conjugated HA trimers per AP205 VLP and ~18–24 trimers per mi3 NP.

## Immunizations with homotypic- and mosaic-HA-VLPs/NPs

Our next goal was to determine whether mosaic HA-VLPs induced a more cross-reactive humoral immune response compared with a mixture of the corresponding homotypic HA-VLPs. We first immunized one group of four mice with mosaic-2 VLPs (presenting CA09 plus Aichi HAs) and a second group of four mice with an equal mixture of CA09-VLPs and Aichi-VLPS (admix-2) (Fig 4A). In addition, we immunized groups of mice with only Aichi-VLPs or only CA09-VLPs. In all cases, mice were primed with equal doses of VLPs plus adjuvant, boosted 2 weeks later without adjuvant, and bled weekly for serum analyses.

Serum ELISAs were performed to measure IgG binding to purified HAs from a panel of group 1 and group 2 influenza A strains (Fig 4B). As expected, IgG titers elicited by immunization with mosaic-2- and admix-2-immunized mice were similar to titers elicited by CA09-VLP immunization against CA09 HA and Aichi-VLP immunization against Aichi-HA. Against heterotypic HAs not coupled tom the VLPs (Viet04, Jp57, WF10, Sh13 and JX346 HAs), IgG titers were similar for both mosaic-2- and admix-2-immunized mice, although titers were consistently higher compared with both CA09-VLP- and Aichi-VLP-immunized mice. Thus in terms of elicited IgG binding of HAs, it appeared that immunizing with the mosaic-2 VLPs that contained group 1 and group 2 HAs was no better at inducing cross-reactive binding of HAs from divergent strains than the corresponding admixture. However, the mosaic-2 and admix-2 injections induced heterologous breadth that could not be explained by the overlapping immunogenicity of the homotypic VLPs.

Next, we determined neutralizing activity of the serum samples using *in vitro* neutralization assays (using infectious viruses for BSL 2 strains and pseudoviruses for BSL 3 strains) against a panel of group 1 and group 2 influenza A strains (Fig 4C). For the mosaic-2- and admix-2-immunized mice, neutralizing titers against homotypic infectious virus strains (CA09 and Aichi) were consistent with the ELISA titers against these strains. Against the Viet04 and JX346 pseudoviruses, neutralization titers were not detectable except for one animal in the admix-2 group. Against the Sh13 pseudovirus, neutralizing titers for the mosaic-2-immunized mice were higher than for the other groups, although the spread in potency was broad and overlapped with the other groups. When considering the neutralization and ELISA results together, the mosaic-2 VLPs did not induce greater breadth than the corresponding mixture of homotypic VLPs.

In order to determine whether mosaic HA-NPs with higher valencies could elicit antibody responses with higher breadth, we conducted experiments similar to those described for VLPs to compare injections of mosaic-2, -4 and -8 NPs with the corresponding admixtures of

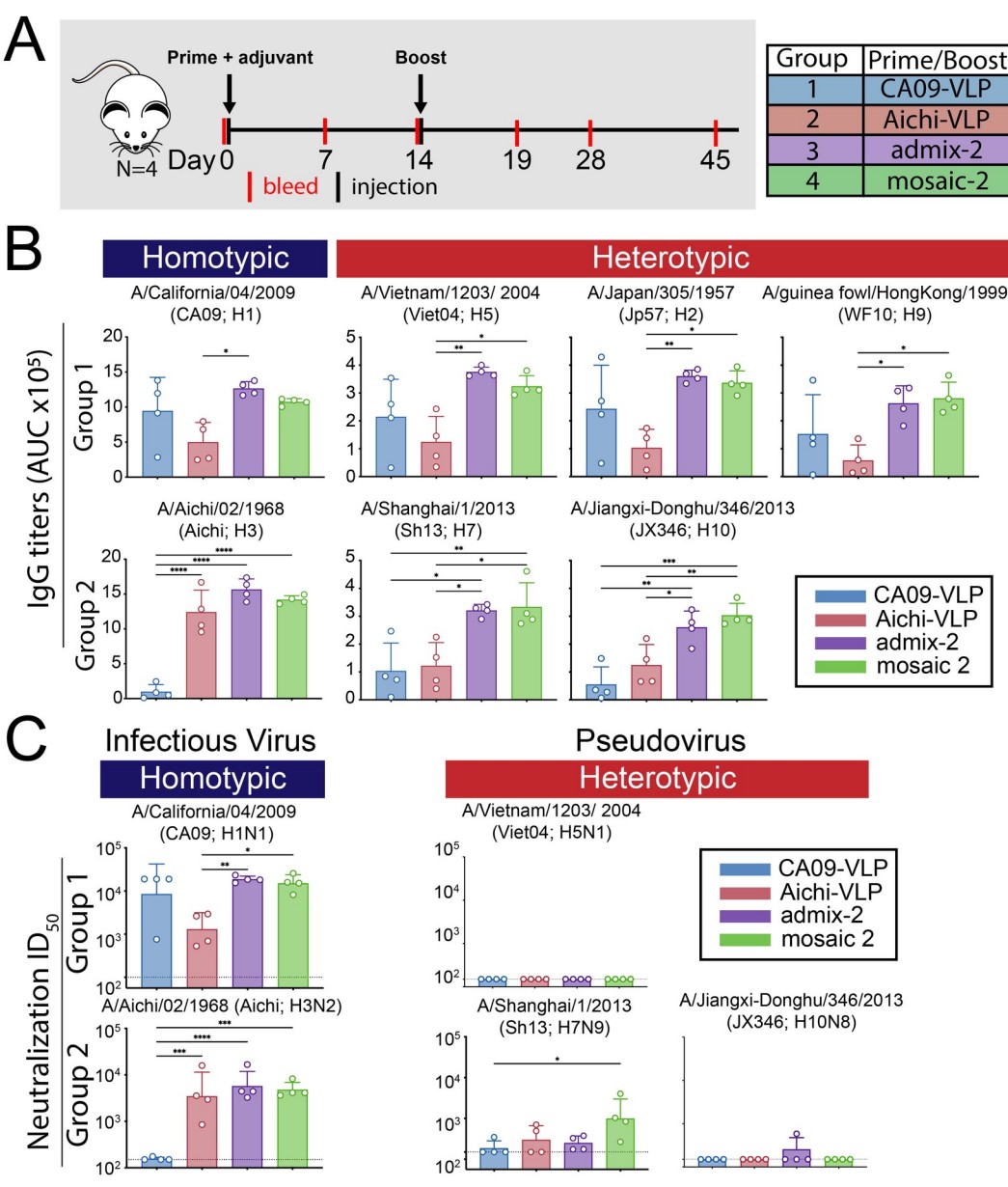

**Fig 4. Immunizations with HA-VLPs. A**. Schematic of the immunization protocol using HA-VLPs (wt HA). Four animals were used for each sample group. **B**. Serum antibody response to wt HA was measured by ELISA and shown as area under the curve (AUC) of Day 28 serum sample to group 1 and group 2 HA trimers. Each dot represents serum from one animal, with arithmetic means and standard deviations represented by rectangles and horizontal lines, respectively. Homotypic strains (present on the mosaic-2 VLP) and heterotypic strains (not present on the mosaic-2 VLP) are indicated by the blue and red rectangles, respectively, above the ELISA data. Significant differences between groups represented by horizontal lines are indicated by asterisks: $p<0.05$ *, $p<0.01$ **, $p<0.001$ ***, $p<0.0001$ ****. **C**. Serum neutralization titers from Day 45 determined by *in vitro* neutralization assays using infectious virus or pseudoviruses. Each dot represents serum from one animal, with geometric mean and geometric standard deviations represented by rectangles and horizontal lines, respectively. Dotted lines indicate limits of detection.

homotypic NPs (Fig 5A), a CA09-NP homotypic control, and unconjugated SpyCatcher-NPs. A final boost was performed 5 weeks after the first prime and animals were sacrificed 2 weeks later to harvest spleens for B-cell analysis. Serum IgG titers from Day 21 and Day 49 were measured by ELISA against a panel of purified HAs from homotypic and heterotypic group1 and

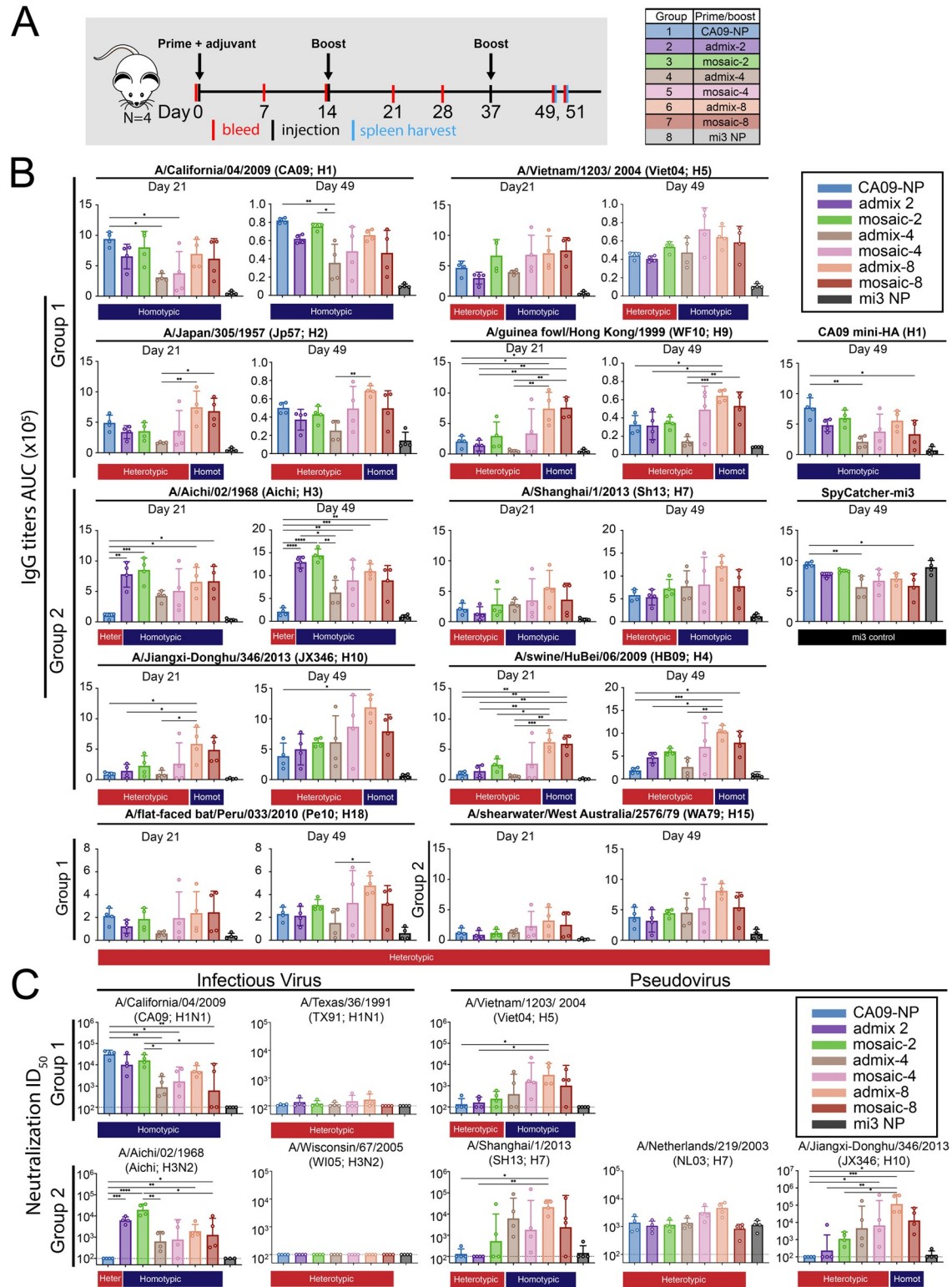

**Fig 5. Immunizations with HA-NPs. A**. Schematic of the immunization protocol using HA-NPs(Y98F). Four animals were used for each sample group. **B**. Serum antibody response to wt HA was tested by ELISA and shown as binding as area under the curve (AUC) of Day 21 and Day 49 serum to recombinant group 1 and group 2 HA trimers. Each dot represents serum from one animal, with means and standard deviations represented by rectangles and horizontal lines, respectively. Homotypic strains that were present on the mosaic NPs and heterotypic strains that were not present are indicated by the blue and red rectangles, respectively, above the

ELISA data. Significant differences between groups represented by horizontal lines are indicated by asterisks: p<0.05 *, p<0.01 **, p<0.001 ***, p<0.0001 ****. **C**. Serum neutralization titers from Day 45 determined by *in vitro* neutralization assays using infectious virus or pseudoviruses. Each dot represents serum from one animal, with geometric means and geometric standard deviations represented by rectangles and horizontal lines, respectively. Dotted lines indicate limits of detection. ND = not determined.

group 2 strains (Fig 5B) and against unconjugated SpyCatcher-NPs (mi3 NPs in Fig 5A). All groups of mice exhibited antibody responses to unconjugated SpyCatcher-mi3, suggesting that SpyCatcher and/or NP epitopes are accessible on HA-conjugated particles.

Against HAs from homotypic strains (CA09 and Aichi), the serum from mice immunized with both the admix and mosaic NPs featured equivalent titers, with the exception of the admix-4 mouse group, which responded with overall lower titers and included 2 mice with no responses. Against the Viet04 and Sh13 HAs (presented on particles with valencies of 4 and 8), the response was slightly higher for the mosaic-2 NPs than for the CA09-NP and admix-2 groups, whereas the mosaic-4, mosaic-8 and admix-8 groups showed higher titers, although over a broad range that included mice with poor responses even against strains of HA that were presented on admix-4 NPs. Against the Jp57, JX346, WF10, and HB09 HAs (only present on the valency-8 NPs), the admix-8 and mosaic-8 titers were equivalent to each other and the highest on average. The titers against heterologous HAs from the CA09-NP-injected mice were similar to titers from mice injected with unconjugated SpyCatcher-NPs, with the exception of responses against Jp57 HA. Responses to the mosaic-2 NPs were slightly higher than responses against admix-2 and CA09-NP, again except for the recognition of Jp57 HA. One animal in the mosaic-4 group exhibited high titers against HAs from all strains in comparison to the admix-4 mice, with the mosaic-4 responses being on par with responses from the animals immunized with the valency-8 NPs. Finally, against HAs from Pe10 and WA79 (not presented on the any of the NPs), serum titers were low for most of the injected animals except for some animals from the mosaic-4 group (e.g., the animal that exhibited high titers against strains not represented on the mosaic-4 NPs), and admix-8- and mosaic-8-immunized mice.

ELISAs were also used to evaluate recognition of CA09-miniHA, a stabilized stem-only construct derived from CA09 HA [11] to investigate whether there was preferred recognition of stem epitopes by the animals immunized with mosaic NPs. We found that the serum response against the CA09 stem was equivalent to responses against the head-containing CA09 HA trimer, with the CA09-NP-immunized mice exhibiting the highest titers (Fig 5B), suggesting that the mosaic NPs did not preferentially elicit anti-stem responses.

ELISA titers determined for serum samples obtained on Day 49 showed similar responses as Day 21 titers with a few exceptions. For example, the mosaic-8 NP-immunized mice had lower titers compared to the admix-8 mice against some of the HAs, suggesting that the additional immunizations were not consistently resulting in strong immune responses. The mice immunized with the mosaic-4 NPs mounted more robust responses, although two of the animals exhibited low titers against all of the strains (as compared with three animals from Day 21). Based on ELISAs, we did not find strong evidence of increased cross-reactivity induced in animals immunized with the mosaic NPs compared with animals injected with the corresponding admix-NPs of the same valency. However, for the mosaic-4 NP group, one animal repeatedly showed high titers of IgG binding to HAs from all strains tested, suggesting that this animal may have induced cross-reactive antibodies.

We also conducted *in vitro* neutralization assays using Day 49 serum against a panel group 1 and group 2 influenza strains (Fig 5C). For CA09, neutralization titers correlated with ELISA titers (S1A Fig), with CA09-NP-immunized mice showing the highest neutralization titers. The neutralizing response to Aichi HA also correlated with ELISA titers (S1C Fig). For the two

heterotypic H1 and H3 HAs (TX91 and WI05 strains), serum from all animals was non-neutralizing, suggesting that neutralizing antibodies that cross-react within the H1 and H3 subtypes were not induced. For Viet04 and Sh13, neutralization correlated with the ELISA titers (S1B and S1D Fig) with valency 4 and 8 mosaic and admix particles showing higher neutralization titers as expected. Against NL03, the neutralizing responses were difficult to interpret due to high background neutralization from the unconjugated mi3 control serum. For JX346, neutralization titers also correlated with ELISA results (S1E Fig). Overall, it appeared that the mosaic NPs did not offer an advantage compared to corresponding mixtures of homotypic particles in induction of neutralizing antibodies, although the mosaic-4 groups included some animals in which greater breadth was induced than admix-4 animals.

## B-cell responses induced by mosaic- versus homotypic-NP immunizations

In order to determine if cross-reactive B-cells were elicited in mice immunized with the mosaic NPs, IgG+ B-cells from immunized mouse spleens were probed for binding to soluble HAs from three different influenza strains using flow cytometry (Fig 6A). The percent binding was determined by gating antigen-specific populations to compare populations positive for CA09, Viet04, or Aichi HAs alone, and for double-positive populations representing B-cells that exhibited cross-reactivity (Fig 6B). As expected, the CA09+ population was the largest for the CA09-NP-immunized mice, with the rest of the mosaic and admix groups eliciting lower proportions of the CA09+ B-cells. Admix and mosaic groups with valencies of 4 and 8 elicited a similar level of Viet04+ IgG+ B-cells, as expected since Viet04 HA was present only on these NPs. Interestingly, the mosaic-2-immunized mice elicited a somewhat lower, but detectable, number of Viet04+ B-cells, which were not present in the admix-2 and homotypic CA09-NP samples. Except for the CA09-NP-immunized mice, all animals showed Aichi+ B-cells; however, the mosaic-2-immunized mice elicited the largest number of antigen-specific B-cells, consistent with ELISA and neutralization results (Fig 6A and 6B). Interestingly, very few CA09 +/Viet04+ B-cells were induced in all immunized animals, with the exception of one animal in the mosaic-4-immunized group, which also featured a high serum IgG and neutralizing response. The CA09-NP and admix-8 immunized mice also induced double-positive B-cells, although to a lesser extent. CA09+/Aichi+ and Viet04+/Aichi+ double-positive B-cells were not detected for any of the animals (data not shown). As expected, antigen-specific B-cell populations (CA09+, Viet04+ and Aichi+) correlated strongly with ELISA serum binding (Aichi-HA, Viet04-HA, and Aichi-HA, respectively; S2A–S2C Fig). Interestingly, the percent of double-positive CA09+Viet04+ B-cells correlated with serum titers for Pe10-HA (S2D Fig), a strain not represented on any of the particles. This suggests that animals that induced CA09+-Viet04+ B-cells also had cross-reactive serum antibodies. Although there was no significant difference in the induction of double-positive B-cells between the mosaic versus admix NP-immunized mice, there is some capacity for the HA-conjugated NPs to induce cross-reactive B-cells and antibodies.

## Comparison of Mosaic NP and VLP immunizations

Several possibilities could account for why no significant differences between mosaic NPs and the corresponding admixture of homotypic NPs were observed. One reason is that there were some animals from each group that did not respond strongly to either prime or boost as determined by ELISA (Fig 5B). Another possibility is that the mi3 NP platform is not as immunogenic as the AP205 VLP platform, which can serve as a self-adjuvant via toll-like receptors since AP205 carries bacterial nucleic acid [18, 19].

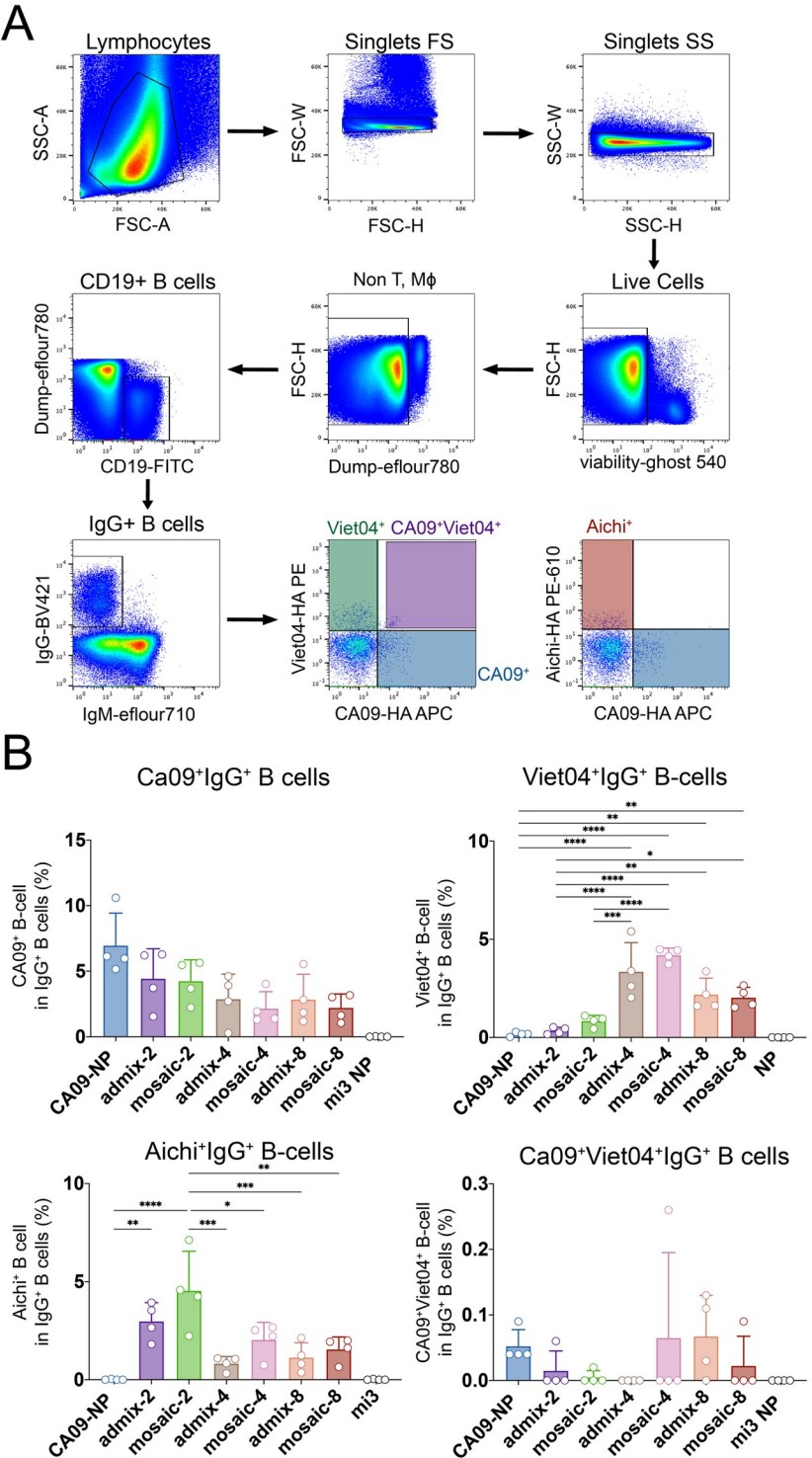

**Fig 6. B-cell responses induced by mosaic NP immunizations. A**. Gating strategy for flow cytometry experiments using single cell suspension from spleens harvested from immunized mice in Fig 5A. Anti-CD3, anti-CD8, anti-F4/80, and anti-Ly6G were used to remove T cells, macrophages, monocytes, and neutrophils. Cells were then gated to isolate CD19/IgG-positive and IgM-negative B-cells, which were probed for binding to CA09-HA-APC (Y98F) (allophycocyanin), Aichi-HA-PE-eFluor610 (Y98F) (phycoerythrin-eFluor 610), and/or Viet04-HA-PE

(phycoerythrin) (Y98F). **B**. Percentage of CA09+, Viet04+, Aichi+, and CA09+/Viet04+ in IgG+ B-cells plotted for each group. Significant differences between groups represented by horizontal lines are indicated by asterisks: $p < 0.05 =$ *, $p < 0.01 = $ **, $p < 0.001 = $ ***, $p < 0.0001 = $ ****. Differences with no significance are not shown, and significant differences between HA-NPs and mi3-NPs are also not shown.

Another animal experiment was conducted to compare mosaic VLPs and mosaic NPs with their counterpart admixtures. Mosaic VLPs and mosaic NPs were prepared with valencies of -4 and -8 along with the corresponding admixtures of homotypic VLPs and NPs (S2A and S2B Fig). Groups of 5 mice were immunized with mosaic and admixture VLPs and NPs, CA09-NP, CA09-VLP, as well as unconjugated NPs and VLPs as controls (Fig 7A). Mice were then boosted with the same antigen in the presence of adjuvant a total of 3 times over the course of 4 months. Mice were bled every two weeks after each immunization. Two weeks after the third boost, mice were sacrificed for B-cell analysis using harvested spleens.

Serum IgG titers were measured via ELISAs using samples from Day 28 against HAs from a panel of group 1 and group 2 strains (Fig 7B). There was no significant difference between antibody titers elicited by mosaic NPs or mosaic VLPs and their counterpart admixtures of equivalent valency against any of the strains tested. Furthermore, there was no major difference between antibody titers from animals immunized with mosaic VLPs versus mosaic NPs. Serum IgG responses against both SpyCatcher-mi3 NPs and SpyCatcher-AP205 VLPs were high not significantly different between all animal groups, suggesting a strong background response against both spycatcher and mi3 or AP205 platforms (S4 Fig).

However, there was a statistically significant difference in ELISA titers against CA09-H1, when comparing mice immunized with CA09-VLPs with respect to mice immunized with mosaic-4 VLPs (p = 0.0018) and mosaic-8 VLPs (p = 0.0084). The difference between CA09 mi3 versus the mosaic NPs was not significant. This suggests that the increase of valency of strains represent on the mosaic VLPs reduced the humoral response against at least one of the component strains.

Somewhat surprisingly, mice immunized with either CA09-NP or CA09-VLP, which only presented the group 1 CA09 H1N1 HA, elicited antibody titers that were cross-reactive against all of the strains tested including the group 2 HAs, often to a similar level as the admix and mosaic VLPs and mosaic NPs that had those strains represented (Fig 7B), suggesting that immunization with a monovalent particle can be sufficient to induce cross-reactivity. Day 42 ELISA titers showed a similar trend (S2B Fig).

## Flow cytometry comparisons of mosaic NP and VLP immunizations

The antigen-specific B-cell response for the mice immunized with mosaic VLPs or mosaic NPs was characterized using flow cytometry (Fig 8A). IgG+ splenocytes were analyzed for binding to a panel of soluble HAs derived from four strains: two from group 1 (CA09 and Viet04) and two from group 2 (Aichi and Sh13). The percent binding was determined for each antigen by gating the antigen-specific CD19+, IgG+, B-cell population that recognized either CA09, Viet04, Sh13, and Aichi HAs alone (Fig 8A), or that recognized double-positive populations that represented cross-reactive B-cells (Fig 8B). Three sets of double-positive cross-reactive antigen specificities were interrogated: CA09+/Viet04+ to look for group 1 breadth, Sh13 +/Aichi+ to look for group 2 breadth, and CA09+/Aichi+ to look for group 1/group 2 breadth (Fig 8B).

Single positive populations for CA09, Viet04, Aichi, or Sh13 correlated with the Day 28 ELISA titers (S5A–S5D Fig). CA09-NP and CA09-VLP elicited a significantly higher percentage of CA09+ B-cells with respect to both the mosaic and admix versions of NPs and VLPs (Fig 8A),

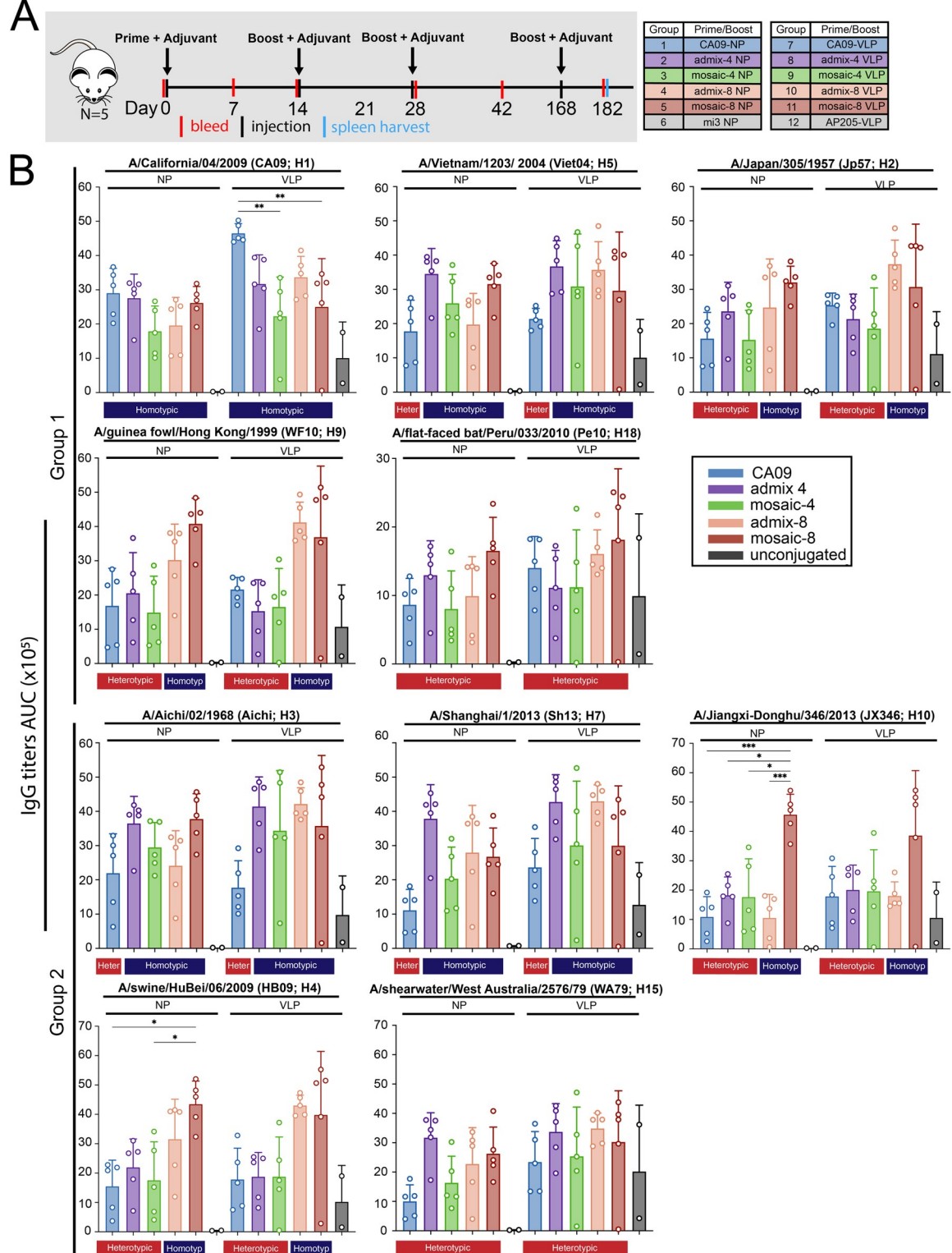

**Fig 7. Immunizations using mosaic VLPs and NPs. A**. Schematic of the immunization protocol (five animals per injection) using HA-NPs (Y98F) and HA-VLPs (Y98F). Mice were immunized in the presence of Addavax adjuvant. **B**. Serum antibody response to wt HA shown by ELISA binding as area under the curve (AUC) of Day 28 serum to recombinant group 1 and group 2 HA trimers. Each dot represents serum from one animal, with means and standard deviations represented by rectangles and horizontal lines, respectively. Homotypic strains that were present on the mosaic NPs and heterotypic strains that were not present are indicated by the blue and red

rectangles, respectively, above the ELISA data. Significant differences between groups represented by horizontal lines are indicated by asterisks: p<0.05 = *, p<0.01 = **, p<0.001 = ***, p<0.0001 = ****. Differences with no significance are not shown, and significant differences between HA-NPs and mi3-NPs, HA-VLPs and Ap205-VLPs, and HA-mi3s and HA-VLPs are also not shown.

in agreement with ELISA results in which the CA09-VLPs elicited higher anti-CA09 titers than mosaic-4 and mosaic-8 VLPs (Fig 7B). There were no statistical significant differences in the percent of antigen-specific B-cells between mosaic-4 and mosaic-8 NPs and VLPs. However, as a general trend, the higher the mosaic valency, the lower the percent of strain-specific B-cells that were elicited, especially in the case of Viet04+ and Aichi+ B-cells (Fig 8A). Interestingly, mice immunized with CA09-NP and CA09-VLP induced antigen-specific B-cells that were specific to Viet04, Aichi, and Sh13 (Fig 8A). This could explain why cross-reactive ELISA titers were observed for these animals against HAs from every strain that was tested.

Similar to the results shown in Fig 6B, induction of cross-reactive CA09+/Viet04+ B-cells were rare (Fig 8B). The difference in the percent of CA09+/Viet04+ B-cells between mosaic, admix, and homotypic VLPs/NPs was therefore not significant. Interestingly, both CA09-NP and Ca09-VLPs were able to induce CA09+/Viet04+ B-cells, suggesting that immunization with monovalent CA09-VLPs/NPs was sufficient to induce cross-reactive B-cells. As previously observed, the percent of CA09+/Viet04+ B-cells correlated with the Day 28 serum ELISA titers against Pe10, a mismatched strain not represented on any of the VLPs or NPs (S5E Fig, p = 0.0234),

Induction of Ca09+/Aichi+ B-cells was observed, although rarely, making it unclear whether they represented B-cells that were cross-reactive to group 1 and group 2 HAs (Fig 8B). Sh13+/Aichi+ B-cells were also observed at a low frequency (Fig 8B). Since both of these populations were rare, there was no significant difference in the percent of double-positive B-cells between each group of immunized mice.

## Conclusions

Attempts to develop broadly protective influenza vaccines have been challenging partly due to the immunodominance hierarchy of antibody epitopes on HA. The variable epitopes on the HA head tend to be more easily recognized than invariant stem epitopes, therefore driving a predominantly strain-specific immune response [2, 3]. A potential strategy to redirect the antibody response towards more conserved stem epitopes is to co-display influenza HAs from different antigenically-distinct strains on particles. A previous study demonstrated the potential for this approach in that antibody responses with greater breadth were observed for mice injected with mosaic HA receptor binding domain particles compared with counterpart homotypic admixtures [16]. Here we sought to extend these results by preparing homotypic and mosaic particles containing trimeric HAs that included stem epitopes that are not present on monomeric HA receptor binding domains. Since HA trimers cannot be fused to ferritin nanoparticles, as previously done to prepare the monomeric HA receptor binding domain particles [16], we used a "plug and display" strategy [18] to covalently couple trimeric HAs to symmetric particles with different numbers of attachment sites (VLPs with 180 attachment sites and NPs with 60 attachment sites), thereby developing a simple method to make homotypic particles displaying a single strain of HA and mosaic particles displaying HAs derived from up to 8 strains. We demonstrated successful conjugation of HA trimers using biochemical methods and EM imaging, including cryo-ET to examine coupling densities of HA on VLPs and NPs. Our biochemical and EM analyses of HA-VLP and HA-NP particles provide useful characterizations for future efforts to utilize the SpyCatcher-SpyTag "Plug and Display" approach [18] for homotypic and heterotypic display of oligomeric antigens.

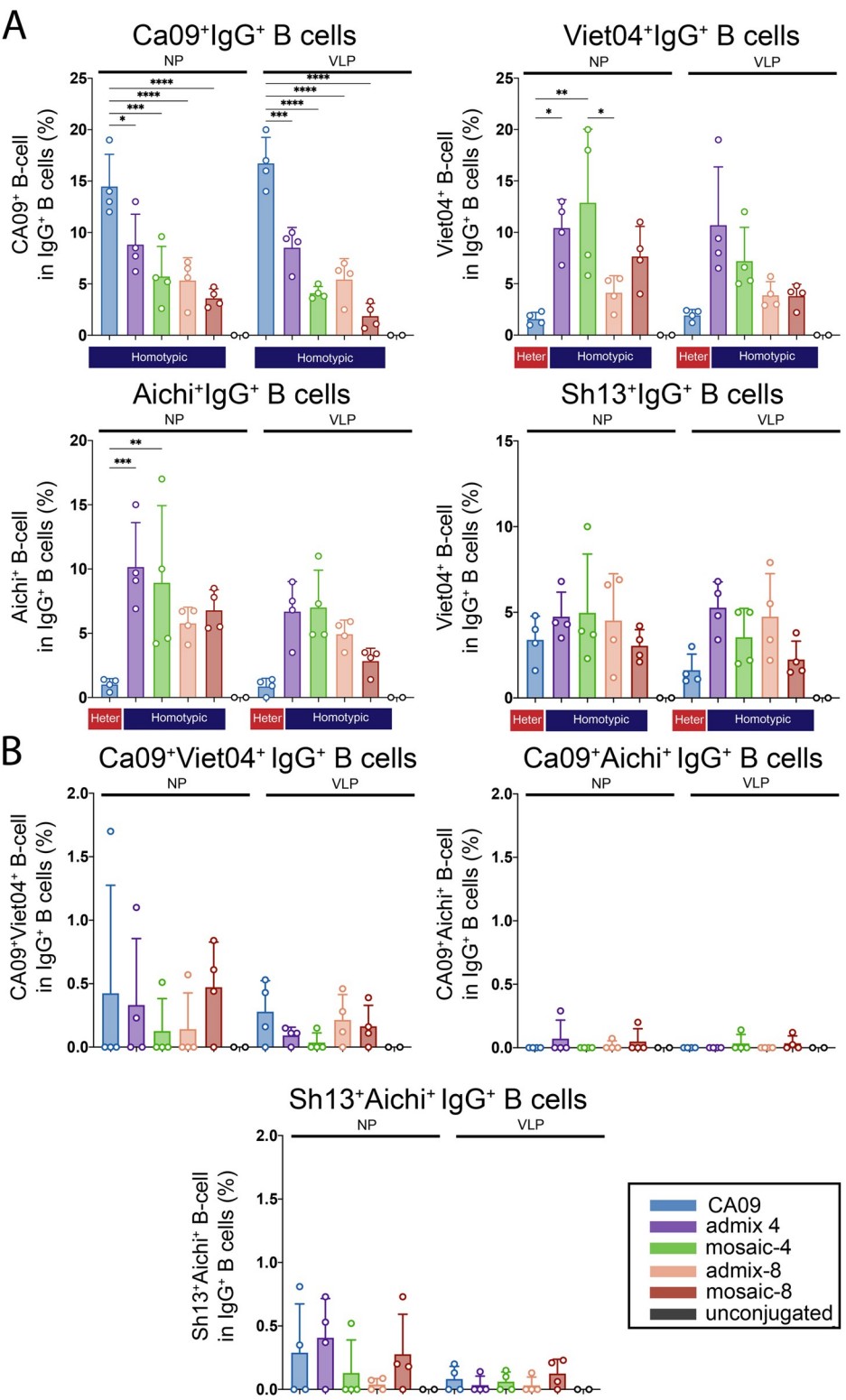

**Fig 8. B-cell responses induced by mosaic VLP and NP immunization.** Flow Cytometry Analysis of IgG+ B-cells isolated from splenocytes as described in **Fig 6A**. **A**. Percent CA09+ (Y98F), Viet04+ (Y98F), Aichi+ (Y98F), and Sh13 + (Y98F) B-cells plotted for each group. **B.** Cross-reactive B-cell compartment: CA09+/Viet04+, CA09+/Aichi+, Sh13 +/Aichi+ B-cell plotted for each group. Significant differences between groups represented by horizontal lines are indicated by asterisks: p<0.05 = *, p<0.01 = **, p<0.001 = ***, p<0.0001 = ****. Differences with no significance are

not shown, and significant differences between HA-NPs and mi3-NPs, HA-VLPs and Ap205-VLPs, and HA-mi3s and HA-VLPs are also not shown.

Our results showed that immunizations with mosaic particles conjugated with HA trimers did not offer a clear advantage in the induction of cross-reactive B-cells compared with immunization of mixtures of homotypic particles. The finding that mosaic particles conjugated with monomeric HA receptor binding domains showed increased induction of cross-reactive B cells compared with admixtures [16], but that mosaic HA trimer particles compared with admix HA trimer particles did not, suggests that potential advantages of mosaic presentation may be related to particular forms of an antigen. For example, the monomeric HA antigens coupled to ferritin were limited to inducing HA head-specific antibodies [16], whereas the trimeric HA ectodomains contained head epitopes as well as stem epitopes that may be partially occluded from interactions with BCRs. In addition, the HA head monomers that were coupled to ferritin were restricted to the H1N1 subfamily [16], whereas our study involved HA trimers derived from group 1 and group 2 influenza strains. Consistent with our results, a recent study using designed nanoparticles to present trimeric HAs from influenza A and B strains also reported no increased breadth of antibody responses against mosaic particles compared with admixtures [29].

Although a clear difference in the degree of cross-reactive B-cells induced by mosaic NPs versus admix NPs when presenting a trimeric HA antigen has not yet been demonstrated, we observed cross-reactive B-cells in response to injections of mosaic and admixture particles. In particular, VLPs and NPs including CA09 HA induced broad responses for both homotypic and heterotypic particles. This suggests the inclusion of CA09 HA antigens on particles in future vaccines. In addition, although mosaic particles and the corresponding admixtures of homotypic particles induced similar levels of increased breadth, the use of mosaic NPs presents a potential therapeutic advantage: i.e., production of a mosaic NP would require purification of one set of particles, whereas use of admix NPs of the same valency would require purification and then mixing of multiple particles prior to immunization. Thus mosaic particles presenting HA antigens derived from multiple influenza strains should be considered as a potential vaccine strategy, and the SpyCatcher-SpyTag "Plug and Display" system [18] can be used to quickly combine different mixtures of oligomeric antigens for preparation of mosaic particles.

## Supporting information

**S1 Fig. Correlation of ELISA AUC titers to live and pseudoviral neutralization titers.** Pearson correlation of Day45 serum ELISA AUC titers and viral neutralization titers for **A.** CA09 H1N1 **B.** Viet04 H5 **C.** Aichi H3N2 **D.** Sh13 H7 **E.** JX346 H10.
(PNG)

**S2 Fig. Correlation of ELISA AUC titers to antigen-specific B-cell populations. A.** Pearson correlation of Day45 CA09+ B-cell population to serum anti-CA09 ELISA AUC titers. **B.** Pearson correlation of Day45 Viet04+ B-cell population to serum anti-Viet04 ELISA AUC titers. **C.** Pearson correlation of Day45 Aichi+ B-cell population to serum anti-Aichi ELISA AUC titers. **B.** Pearson correlation of Day45 CA09+Viet04+ B-cell population to serum anti-Pe10 ELISA AUC titers.
(PNG)

**S3 Fig. Conjugation of SpyCatcher-VLPs and -NPs. A**. SpyCatcher-AP205-VLP and Spy-Catcher-mi3 conjugations with SpyTagged-HA trimers. **B**. Purification of homotypic and mosaic SpyCatcher-VLPs and SpyCatcher-mi3s. Left: SEC separation of conjugated NPs from

free HA trimers. Right: Reducing SDS-PAGE analysis of NPs and purified HAs.
(TIFF)

**S4 Fig. Immunizations using mosaic VLPs and NPs.** Serum antibody response to HA shown by ELISA binding as area under the curve (AUC) of Day 14 serum to SpyCatcher NP and VLP particles, with means and standard deviations represented by rectangles and horizontal lines, respectively. Homotypic strains that were present on the mosaic NPs and heterotypic strains that were not present are indicated by the blue and red rectangles, respectively, above the ELISA data.
(PNG)

**S5 Fig. Correlation of ELISA AUC titers to antigen-specific B-cell populations. A.** Pearson correlation of Day28 CA09+ B-cell population to serum anti-CA09 ELISA AUC titers. **B.** Pearson correlation of Day28 Aichi+ B-cell population to serum anti-Aichi ELISA AUC titers. **C.** Pearson correlation of Day28 Viet04+ B-cell population to serum anti-Viet04 ELISA AUC titers. **D** Pearson correlation of Day28 Sh13+ B-cell population to serum anti-Sh13 ELISA AUC titers. **E.** Pearson correlation of Day45 CA09+Viet04+ B-cell population to serum anti-Pe10 ELISA AUC titers.
(PNG)

**S1 Checklist.**
(PDF)

**S1 Raw images.**
(ZIP)

**S1 Movie.**
(MP4)

**S2 Movie.**
(MP4)

## Acknowledgments

We thank Mark Howarth (Oxford University) for providing plasmids and advice for VLP expression and purification, Jesse Bloom (Fred Hutchinson) for reagents for the infectious virus neutralization assays, Jost Vielmetter and Pauline Hoffmann at the Caltech Beckman Institute Protein Expression Center for help with protein production, Andrey Malyutin and Songye Chen (Caltech) for help with cryo-EM data collection, Rochelle Diamond and Jamie Tijerina at the Caltech Flow Cytometry/Cell Sorting Facility for help in the flow cytometry experiments and analysis, Jennifer Keeffe for implementation of influenza neutralization assays, Claudia Jette for help with figure preparation, Harry Gristick for VLP and NP images for figures, and Andrew Flyak, Jennifer Keeffe, and Claudia Jette for critical reading of the manuscript. EM was done in the Beckman Institute Resource Center for Transmission Electron Microscopy at Caltech.

## Author Contributions

**Conceptualization:** Alexander A. Cohen, Zhi Yang, Pamela J. Bjorkman.

**Data curation:** Alexander A. Cohen.

**Formal analysis:** Alexander A. Cohen, Zhi Yang, Pamela J. Bjorkman.

**Investigation:** Alexander A. Cohen.

**Methodology:** Alexander A. Cohen, Zhi Yang, Priyanthi N. P. Gnanapragasam, Susan Ou, Kim-Marie A. Dam, Haoqing Wang.

**Project administration:** Alexander A. Cohen, Priyanthi N. P. Gnanapragasam, Susan Ou, Kim-Marie A. Dam.

**Resources:** Alexander A. Cohen.

**Supervision:** Pamela J. Bjorkman.

**Visualization:** Alexander A. Cohen.

**Writing – original draft:** Alexander A. Cohen, Pamela J. Bjorkman.

**Writing – review & editing:** Alexander A. Cohen, Pamela J. Bjorkman.

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
