## [Decision Letter · Decision Letter 0]

24 Dec 2020

PONE-D-20-28839

Construction, characterization, and immunization of nanoparticles that display a diverse array of influenza HA trimers

PLOS ONE

Dear Dr. Bjorkman,

Thank you for submitting your manuscript to PLOS ONE. After careful consideration, we feel that it has merit but does not fully meet PLOS ONE’s publication criteria as it currently stands. Therefore, we invite you to submit a revised version of the manuscript that addresses the points raised during the review process.

During the revision process, please address the issues related to choice of vaccine delivery route and provide additional insight into the potential responses in the germinal center and at the T cell level.

We look forward to receiving your revised manuscript.

Kind regards,

Victor C Huber

Academic Editor

PLOS ONE

Journal Requirements:

2.) As part of your revision, please complete and submit a copy of the ARRIVE Guidelines checklist, a document that aims to improve experimental reporting and reproducibility of animal studies for purposes of post-publication data analysis and reproducibility: https://www.nc3rs.org.uk/arrive-guidelines. Please include your completed checklist as a Supporting Information file. Note that if your paper is accepted for publication, this checklist will be published as part of your article.

3.) PLOS ONE now requires that authors provide the original uncropped and unadjusted images underlying all blot or gel results reported in a submission’s figures or Supporting Information files. This policy and the journal’s other requirements for blot/gel reporting and figure preparation are described in detail at https://journals.plos.org/plosone/s/figures#loc-blot-and-gel-reporting-requirements and https://journals.plos.org/plosone/s/figures#loc-preparing-figures-from-image-files. When you submit your revised manuscript, please ensure that your figures adhere fully to these guidelines and provide the original underlying images for all blot or gel data reported in your submission. See the following link for instructions on providing the original image data: https://journals.plos.org/plosone/s/figures#loc-original-images-for-blots-and-gels.

4.) We note that you have included the phrase “data not shown” in your manuscript. Unfortunately, this does not meet our data sharing requirements. PLOS does not permit references to inaccessible data. We require that authors provide all relevant data within the paper, Supporting Information files, or in an acceptable, public repository. Please add a citation to support this phrase or upload the data that corresponds with these findings to a stable repository (such as Figshare or Dryad) and provide and URLs, DOIs, or accession numbers that may be used to access these data. Or, if the data are not a core part of the research being presented in your study, we ask that you remove the phrase that refers to these data.

Reviewers' comments:

Reviewer's Responses to Questions

**Comments to the Author**

1. Is the manuscript technically sound, and do the data support the conclusions?

Reviewer #1: Yes

Reviewer #2: Yes

2. Has the statistical analysis been performed appropriately and rigorously? 

Reviewer #1: Yes

Reviewer #2: Yes

3. Have the authors made all data underlying the findings in their manuscript fully available?

Reviewer #1: Yes

Reviewer #2: Yes

4. Is the manuscript presented in an intelligible fashion and written in standard English?

Reviewer #1: Yes

Reviewer #2: Yes

5. Review Comments to the Author

Reviewer #1: This excellent manuscript by Cohen et al. describes nice “Plug and Display” virus like particle (VLP) and nanoparticle (NP) vaccine platforms to present trimeric HA proteins. Both platforms are immunogenic and allow quick generation of mosaic particles displaying HAs of different strains, while the NP platform shows improved yields. The group was able to generate VLPs and NP with 2, 4 and 8 HA valency aiming to target the cross-reactive epitopes on different HA strains. As truthfully reported by the authors, this mosaic strategy did not successfully induce cross-reactive antibodies or B-cells. This, on the other hand, seems to support the hypothesis that an immunodominant hierarchy hinders the induction of cross-reactive antibodies, in which the variable epitopes (typically in the globular head domain) are more accessible than the conserved epitopes in the stem by the BCR, regardless of number of epitopes that are presented at the same time. Nevertheless, this study shows promising vaccine platforms that might be more advantageous when used as an improved seasonal vaccine instead of a universal influenza virus vaccine.

1. The HA-VLP in Figure 7 appears to elicit higher antibody responses. The authors hypothesized that this could due to the self-adjuvant effect of the VLPs. However, Page 14, line 2 “The HAs for the SpyCatcher-mi3 conjugations include the sialic acid binding knockout mutation Y98F”; This gives the impression that HAs conjugated with VLP do not have the Y98F mutation. If that is the case, ELISAs performed using recombinant HA without the Y98F could give different results for the two platforms. The authors should provide more clarification of whether the wt or mutant HA were used in the VLP platform and the ELISAs.

2. Is the SpyCatcher immunogenic? Have the author measured antibodies specific to the Spycatcher or the NP/VLP separately?

3. The immunization route and adjuvant used in Fig 4 and 5 are different from those used in Fig 7. Can the author explain a little bit more why that is?

Reviewer #2: The manuscript by Cohen et al. demonstrates that in two different nanoparticle formulations (AP205 VLPs and mi3) carrying multiple strains of influenza HA, there are no major differences in antibody responses between antigen (HA trimers) presented as mosaic particles or admixed particles. Although the results are "negative" per se, the manuscript provides valuable insights for the field and kudos to the authors for such a through work. However, prior to publications several questions should be addressed and some new data needs to presented.

1. The choice of i.p delivery of the vaccine in mice is interesting but unusual. Why was i.p. chosen as the route? Typically intramuscular, intradermal/subcutaneous, or intranasal routes are chosen.

2. Please provide statistics in the figures. There are discussions of the stats in the text, but at least each figure or figure legend should summarize the statistics for the corresponding data.

3. Do the authors have any data on the germinal center reaction for these groups? It is critical to understand, from an Ab response point of view, whether the GC responses were all similar or whether these formulations failed to elicit strong GC responses. This would provide mechanistic insights better than just looking at Ab responses.

4. It is disappointing that no T cell response data were shown. It would much to the study if lymph nod eor splenic T cell assays are performed and reported.

6. PLOS authors have the option to publish the peer review history of their article (what does this mean?). If published, this will include your full peer review and any attached files.

Reviewer #1: No

Reviewer #2: No

---

## [Author Response · Author response to Decision Letter 0]

13 Feb 2021

1. The choice of i.p delivery of the vaccine in mice is interesting but unusual. Why was i.p. chosen as the route? Typically intramuscular, intradermal/subcutaneous, or intranasal routes are chosen.

The IP immunization route was necessary to deliver the desired volume for each injection.

3. Do the authors have any data on the germinal center reaction for these groups? It is critical to understand, from an Ab response point of view, whether the GC responses were all similar or whether these formulations failed to elicit strong GC responses. This would provide mechanistic insights better than just looking at Ab responses.

We agree with the reviewer that it would be very interesting to examine GC responses. Unfortunately, we do not have appropriate samples from the immunized mice at this time, so doing these experiments would require repeating the entire immunization series in mice, which would be both expensive and time-consuming. We note that the other papers examining responses to mosaic HA nanoparticles that we cited (Kanekiyo et al., 2019 and Boyoglu-Barnum et al., 2020) did not include GC studies, thus suggesting that these experiments are beyond the scope of our paper and the previous studies.

4. It is disappointing that no T cell response data were shown. It would much to the study if lymph nod eor splenic T cell assays are performed and reported.

Again, we agree with the reviewer that it would be interesting to examine T cell responses. However, we no longer have lymph node or spleen samples from the immunized mice, and note that T cell assays were not reported in other papers reporting mosaic HA nanoparticles (Kanekiyo et al., 2019 and Boyoglu-Barnum et al., 2020).

---

## [Editor Report · Decision Letter 1]

17 Feb 2021

Construction, characterization, and immunization of nanoparticles that display a diverse array of influenza HA trimers

PONE-D-20-28839R1

Dear Dr. Bjorkman,

We’re pleased to inform you that your manuscript has been judged scientifically suitable for publication and will be formally accepted for publication once it meets all outstanding technical requirements.

Kind regards,

Victor C Huber

Academic Editor

PLOS ONE
---

## [Editor Report · Acceptance letter]

23 Feb 2021

PONE-D-20-28839R1 

Construction, characterization, and immunization of nanoparticles that display a diverse array of influenza HA trimers 

Dear Dr. Bjorkman:

I'm pleased to inform you that your manuscript has been deemed suitable for publication in PLOS ONE. Congratulations! Your manuscript is now with our production department. 

Kind regards, 

on behalf of

Dr. Victor C Huber 

Academic Editor

PLOS ONE